# Pulse Spray Drying for Bovine Skimmed Milk Powder Production

**DOI:** 10.3390/foods13060869

**Published:** 2024-03-13

**Authors:** María Romo, Doll Chutani, Dinar Fartdinov, Ram Raj Panthi, Nooshin Vahedikia, Massimo Castellari, Xavier Felipe, Eoin G. Murphy

**Affiliations:** 1Food Processing and Engineering Programme, Institute for Food and Agricultural Research and Technology (IRTA), Granja Camps i Armet s/n, Monells, 17121 Girona, Spain; maria.romo@irta.cat (M.R.); dinar.fartdinov@irta.cat (D.F.); 2Food Chemistry and Technology Department, Moorepark Food Research Centre, Teagasc, Fermoy, Co., P61 C996 Cork, Ireland; doll.chutani@teagasc.ie (D.C.); ram.raj.panthi@gmail.com (R.R.P.); nooshin.vahedikia@teagasc.ie (N.V.); eoin.murphy@teagasc.ie (E.G.M.); 3Functional and Food Safety Programme, Institute for Food and Agricultural Research and Technology (IRTA), Granja Camps i Armet s/n, Monells, 17121 Girona, Spain; massimo.castellari@irta.cat

**Keywords:** SMP, Pulse Combustion Drying, Spray Drying, native whey proteins, microstructure

## Abstract

Pulse Spray Drying (PSD) has potential as a sustainable means of skimmed milk powder (SMP) production. In this study, powders were obtained from PSD using different drying outlet temperatures (70, 80, 90 and 100 °C), and their characteristics were compared to those of traditional Spray Drying (SD). Native whey proteins were well preserved and Solubility Indexes were over 98% in all cases, despite powders obtained by PSD displaying lower solubility than the SD ones. No visual difference was observable in the powders (ΔE < 2); however, PSD powders were found to be yellower with a higher Browning Index. The drying technology did not have a significant effect on powder moisture content and bulk density. Particle size distribution and scanning electron microscopy images confirmed the presence of fine particles (<10 μm) in all samples that might have provided poor flowability and wetting behavior (overall Carr Index and Hausner ratio were 33.86 ± 3.25% and 1.52 ± 0.07, respectively). Higher amounts of agglomerated particles were found at low temperatures in the products processed with both technologies, but PSD samples showed a narrower particle size distribution and hollow particles with more wrinkles on the surface (probably due to the fast evaporation rate in PSD). Overall, PSD provided SMP with comparable physicochemical characteristics to SD and, once optimized at the industrial level, could offer significant advantages in terms of thermal efficiency without significant modification of the final product quality.

## 1. Introduction

Drying increases the shelf-life of perishable products by considerably reducing the water content, which also facilitates transportation to far-away markets. Skimmed milk powder (SMP) is widely used in the food industry (cheesemaking, infant formulation, confectionary, ice cream and desserts, beverages or bakery) and is mostly obtained by Spray Drying [1]. Spray Drying (SD) is an effective, simple, highly versatile and fast technique that allows for the continuous production of powders. Liquid feed is exposed to a flow of drying gas, mostly hot air, and due to the short residence time of the product in the drying chamber (1–10 s), SD is suitable for thermolabile compounds [2,3,4]. Even though SD is applied in a large variety of industries (food powders, nutraceuticals, probiotics, enzymes and antibiotics), it has been described as a high-energy and low-heat efficiency technique [5,6]. Furthermore, SD is not suitable for processing highly viscous liquids because of the limitations of the atomization step, performed with nozzles or rotary wheels/disks [7,8]. In response to the increased demand for more sustainable technologies in food processing, research into the development of lower-energy drying is currently ongoing, including research relating to variants or modifications of the traditional SD for powder manufacture [9].

Pulse Spray Drying (PSD), proceeding from pulse combustion first discovered by Higgins in 1777 [10], has been widely developed but not until the 1970s it was evaluated for different applications such as food processing [9]. The PSD mechanism has been thoroughly described in the literature and the Helmholtz pulse combustor type is the most frequently employed in industry [9,11,12,13]. First, a mixture of fuel and air enters the chamber from an inlet valve where it is ignited, leading to an increase in pressure that closes the valve. The hot combustion gases expand outward to the tail pipe and the pressure decreases, sucking inward some gases and opening the valve again. Fresh mixture (air and fuel) enters and is spontaneously ignited when it comes into contact with the remaining combustion products of the preceding cycle. All of this results in a sonic wave that leaves the combustion chamber through a tailpipe, where it comes into contact with the product to be dried. The pulses associated with the combustion then break the feed product into fine droplets for subsequent drying. The performance of PSD depends on the supply rate of the mixture fuel/air, the nature of the fuel, feed product characteristics and tailpipe length [14,15,16].

PSD is reported to offer several advantages over the continuous combustion systems:(a)Mass transfer and heat rates are intensified, with an increase of up to 40% reported for thermal efficiency [17,18];(b)Feed residence time is lower (milliseconds), allowing the application of higher temperatures; reduced contaminant emissions (NOx, CO and SOx) [11,19,20];(c)It is appropriate for heat-sensitive and viscous compounds and reduced space required for the equipment [15].

PSD has been successfully tested on different materials such as drugs [21,22], powder based on zirconium and zinc oxide [23,24], wood and thick-grade paper [25,26], food wastes [27], egg white [28] or maltose solutions [29].

During the drying process, high temperatures can cause the denaturation of whey proteins [30] and can induce alterations to the physical and functional properties of the resulting powder (e.g., particle structure, particle size distribution, flowability, bulk density or solubility). Egg white powder obtained by PSD (76.6 °C of outlet drying temperature) has been reported to present finer particles, in agreement with other studies [13,24,31,32], with whiter color and superior surface characteristics than an equivalent SD powder dried under similar drying conditions [28]. However, studies detailing the application of PSD to food matrices are very limited and, to the best of our knowledge, no study has been published on skim milk. Therefore, this study aimed to bridge this gap by comparing from physicochemical properties of skimmed milk powder (SMP) obtained by PSD to SD obtained using a number of different outlet drying temperatures (70, 80, 90 and 100 °C).

## 2. Materials and Methods

### 2.1. Chemicals and Standards

Acetonitrile HPLC grade (ACN), trifluoracetic acid (TFA), sodium acetate, acetic acid and silicon antifoam were provided by Sigma-Aldrich (Sigma-Aldrich^®^ Merck, Darmstadt, Germany). Reference standards of bovine whey proteins (α-La and β-Lg A and B isoforms) were provided by Cerilliant (Sigma-Aldrich^®^ Merck, Darmstadt, Germany).

### 2.2. Milk Sample Preparation

Eight batches of bovine pasteurized skim milk (HTST) were provided by Lactics Tramuntana (Girona, Spain) and transported to the pilot plant at 4 °C at different times over an eight-week period. Milk was treated following manufacturer recommendations [33]. Each batch was preheated (65 ± 2 °C) and evaporated up to 28.81 ± 0.67% of total solids (*w*/*w*) using a falling film evaporator (evaporation capacity of 100 kg/h and 200 mbar vacuum; FF200, GEA Niro, Montigny le Bretonneux, France) at 73 ± 2 °C. The skim milk concentrate (SMC) (pH 6.3; 0.44 ± 0.19% *w*/*w*, 10.29 ± 0.26% *w*/*w* and 2.35 ± 0.05% *w*/*w* of fat, protein and ash, respectively) was cooled to 4 °C and stored until drying (<18 h).

### 2.3. Proximate Composition Parameters

Fat, protein content, total solids and ash were measured by ISO 1211/IDF [34], ISO 8968-3/IDF20-3 [35], ISO 673/IDF21 [36] and B.O.E. O.M. 31/01/77 [37], respectively. The protein content of the powders was determined using LECO Nitrogen Analyser FP-638 (LECO Corporation, St Joseph, MI, USA), with a nitrogen-to-protein conversion factor of 6.38.

### 2.4. Drying Process

Each batch of SMC was divided into two lots as specified below. The process was repeated twice for each temperature condition:Spray Drying (SD) pilot model (evaporation capacity of 2.2 kg/h; Minor Mobile, GEA Niro, Denmark). Four air temperature conditions were studied (70, 80, 90 and 100 °C outlet temperatures with the respective inlet temperatures of 160, 180, 190 and 210 °C). The feed (flow rate: 48.3 g/min) was atomized by using compressed air at 3 bar filtered at 5 µm (AC20-F02G SMC, Amidata, Madrid, Spain).Pulse Spray Drying (PSD) pilot model (evaporation capacity of 70 kg/h; PSD-70; Ekonek, Spain). The different outlet temperatures (70, 80, 90 and 100 °C) were adjusted by applying variable flow rates of feed (72, 62, 52 and 42 L/h, respectively). The feed was dispersed by the internal pressure wave generated in the combustion motor at 148 Hz and a constant inlet flow of propane (4.7 kg/h) was applied. The product was collected in polypropylene boxes located under the rotary valves under the main chamber and the end of the cyclone separator, and mixed at 50/50 (*w*/*w*) in a fluidized bed (Strea 1-Pro, Gea Niro, Montigny le Bretonneux, Denmark) for 10 min (constant air flowrate of 70 ± 2 kg·h^−1^).

The powders obtained at each condition were kept in high-density polyethylene bottles (Lamaplast srl, Sesto san Giovanni (MI), Italy) that were packed in aluminum bags under vacuum until analyses.

### 2.5. RP-HPLC Quantification of Soluble Native Whey Proteins

Reverse-phase high-performance liquid chromatography (RP-HPLC) was used for soluble native whey protein quantification. Powders were reconstituted in ultrapure water up to a protein content of 3% overnight under stirring (4 °C). Reconstituted SMPs were acidified and diluted (1:1) in a 1 M sodium acetate buffer (pH 4.6). Samples were centrifuged (20,000 rpm, 20 min, 4 °C; Eppendorf 5417R, Hamburg, Germany) and the supernatant was filtered directly into the vials through Econofltr PES filters (25 mm diameter, 0.2 μm pore size; Agilent Technologies, Santa Clara, CA, USA).

Protein separation was performed with an Agilent Technologies 1200 Series (Santa Clara, CA, USA) equipment on a ZORBAX 300 SB-C18 StableBond Analytical column (150 × 4.6 mm; 5-Micron; Agilent Technologies, Santa Clara, CA, USA) at 40 °C. A total of 15 μL of each sample were injected twice and the elution was performed in gradient conditions (0–10 min 18–25% B, 10–12 min 25–31.5% B, 12–15 min 31.5–38.5% B, 15–20 min 38.5–40% B, 20–25 min 40–44% B, 25–32.5 min 44–52% B, 32.5–35 min 52–90% B., 35–38 min 90% B and 38–45 min 18% B) at a flowrate of 1 mL·min^−1^ with a binary mobile phase (A: 0.1% TFA in water; B: 0.1% TFA in ACN). Peaks were detected at 214 nm and pure bovine α-La and β-Lg A + B commercial standards were used for identification and concentration calculations.

### 2.6. Powder Physical Properties

Solubility Index (S_I_) (%) was calculated according to Equation (1), where V_i_ stands for the initial volume of the sample and I_I_ (Insolubility Index) was obtained following ISO 8156:2005(E) [38]. Wettability was calculated as the necessary time (s) for a determined amount of powder to become wet following the GEA Niro (2012) protocol [39]. GEA Niro (2012) protocols were also followed for obtaining powder moisture content and bulk volume. Bulk density was calculated from the bulk volume [39]. Compressibility (or Carr Index) of the powder was given by the difference between poured (ρ_bulk_) and tapped (1250 times) (ρ_tapped_) bulk densities following Equation (2) [40]. Hausner ratio was calculated as ρ_tapped_/ρ_bulk_ following Hausner et al. [41].
(1)$$\mathrm{Solubility}\;\mathrm{Index}\;\left(\%\right)=\frac{V_{i}-{\;I}_{I}}{V_{i}}\cdot 100$$
(2)$$\mathrm{Compressibility}\;\left(\%\right)=\frac{\rho_{\mathrm{tapped}}-\rho_{\mathrm{bulk}}}{\rho_{\mathrm{tapped}}}\cdot 100$$

Flowability was determined according to GEA Niro (2012) [39], by measuring the seconds a given volume of powder takes to leave a rotary drum. Results (g/min) are calculated as (P_i_ − P_f_)/time, where P_i_ and P_f_ are the weights before and after the test, respectively. Flowability was classified according to the compressibility and Hausner ratio described by Lebrun et al. [42].

A dynamic vapor sorption analyzer DVS-1 (Surface Measurement Systems Ltd., London, UK) equipped with a Cahn microbalance was used to obtain sorption isotherms. Relative humidity (RH) was adjusted from 0 to 90% at 25 °C. Samples (10 mg) were first dehydrated (RH ≈ 0%) and then hydrated with 10% RH steps. Two replicates per treatment condition were analyzed.

### 2.7. Particle Size and Distribution

Particle size distribution of the SMP obtained by SD and PSD was determined by laser light scattering using a Mastersizer 3000 (Malvern Instruments Ltd., Worcestershire, UK). The refractive index and absorption coefficient used were 1.46 and 0.001, respectively [43]. Sauter mean diameter (D_[3,2]_), volume mean diameter (D_[4,3]_), D_10_, D_50_, D_90_ and span of particles [(D_90_ − D_10_)/D_50_] were measured.

### 2.8. Colour Measurements

A portable colorimeter Minolta Chroma Meter CR-400 (Konica Minolta, Tokyo, Japan) was used for studying the SMP obtained by PSD and SD. Each measurement was performed three times and results were expressed in *L** (Lightness), *a** (redness-greenness) and *b** (yellowness-blueness). Browning Index (BI) was calculated according to Al-Hilphy et al. [44] using Equation (3), where x is (*a** + 1.75L*)/(5.645L* + *a** − 3.012*b**).

Color differences (Δ*E*) of the samples were calculated with Equation (4), where *L*_0_*, *a*_0_* and *b*_0_* are the values corresponding to the powder obtained by SD under the same drying temperature condition.
(3)$$\mathrm{Browning}\;\mathrm{Index}\;\left(\mathrm{BI}\right)=\left(100\cdot \left(x-0.31\right)\right)/0.172$$
(4)$$\Delta E=\sqrt{{\left(L_{0}^{*}-L^{*}\right)}^{2}+{\left(a_{0}^{*}-a^{*}\right)}^{2}+{\left(b_{0}^{*}-b^{*}\right)}^{2}})$$

### 2.9. Scanning Electron Microscopy (SEM)

The structure of the SMP was visualized by field-emission scanning electronic microscopy (SEM) (Zeiss Supra 40VP with a Gemini Column, Carl Zeiss AG, Oberkochen, Germany). Samples were placed on double-sided carbon tape mounted on SEM stubs and coated with gold ions in a sputter coater (Emitech K575X, Quorum Technologies, East Sussex, UK). A range of 50 to 5000 magnification was studied.

### 2.10. Statistics

Statistical analyses were performed with JMP^®^ software version 17.2.0 (SAS Institute Inc., Cary, NC, USA). Variance tests (ANOVA) were assessed with a significant level set at *p* < 0.05 and significant effects of the independent variables on the dependent variables were studied by Tukey’s Honest Significant Difference (HSD) (alpha = 0.05).

## 3. Results and Discussion

The effect of drying temperature (70–100 °C) and technology (PSD-SD) on the physicochemical properties of SMP is shown in Appendix A.

### 3.1. Protein and Moisture Content

Suitable moisture content in skimmed milk powders (SMPs) is crucial to preserve quality; low values may lead to oxidation, while high values can affect a number of important properties (e.g., protein denaturation, Maillard reactions, caking or microbial growth) [45]. In the current study, both drying technology and temperature had a significant effect on moisture content for all SMPs (*p* = 0.034 and *p* = 0.0002, respectively). Powders processed by PSD were less humid than by SD (2.37 ± 1.19 and 2.95 ± 0.89%, respectively) (Table 1), but in all cases, moisture content values were in the range described for these products [46]. The overall effect of the outlet drying temperature on SMPs is shown in Table 2. As expected, higher drying temperatures could remove the water more efficiently, leading to a lower moisture content of the final product [47,48].

Several studies have demonstrated that milk proteins can suffer structural modifications, mostly due to heat exposure, during drying (e.g., glycation, phosphorylation, unfolding and denaturation) [49,50]. In the current work, total protein content was, in all cases, in the range of 33–36%, in agreement with literature for SMPs [51,52], and increased at high temperatures in samples processed with both technologies (Figure 1). The application of high temperatures during the drying led to more water removal (Table 2), which could have provoked an increment in the concentrations of powder components, explaining the increase in total protein content. α-Lactalbumin (α-La) and β-Lactoglobulin (β-Lg) were quantified in order to study the effect of SD and PSD on soluble whey proteins. The drying technology and temperature did not have a significant effect on the concentration of α-La and β-Lg in the final product, which ranged between 8.30–13.38 and 39.32–61.00 g/Kg, respectively (Appendix A). These results are in agreement with Oldfield et al. [53], who described minimal denaturation of whey proteins during SD at 89 and 100 °C outlet temperatures.

PSD successfully provided SMPs with comparable moisture and total protein content values, and minimal whey protein denaturation, in comparison with SD.

### 3.2. Color

The color of dairy powders is an essential factor for consumer/customer acceptability of the final product. When studying color differences between PSD and SD samples, the drying temperature did not have a significant effect on luminosity (*L**), redness (*a**), yellowness (*b**) or Browning Index (BI). However, PSD powders showed yellower tonalities (*p* = 0.0307), displaying an overall higher BI (*p* = 0.0052) (*b** = 10.00 ± 0.37; BI = 15.09 ± 0.77) than SD samples (*b** = 9.02 ± 0.91; BI = 12.82 ± 5.03) (Table 1). Generally, skimmed milk powders are rich in lactose, which can trigger Maillard reaction during treatments at high temperatures [54,55], which has been associated with an increase in BI [56]. Thus, further studies would be required to evaluate the actual extent of Maillard reaction in PSD samples. Notwithstanding, the overall effect of PSD application on Δ*E* was less than 2 units (Δ*E* = 1.16) when compared to SD, which is in keeping with the anecdotal observation that the differences mentioned above were not appreciable to the human eye [57].

### 3.3. Particle Size and Distribution

The particle size distribution of SMP obtained by PSD and SD is shown in Figure 2 and Table 3.

Temperature and drying technology had a significant effect (*p* < 0.0001) on particle size parameters for all SMPs. Particle size distribution profiles of skimmed milk powders from SD and PSD were remarkably different. The highest span value, which indicates a wider distribution [58], was obtained in powders dried by SD at 70 °C (4.52), while PSD profiles became narrower with the temperature increment.

Samples obtained with SD displayed a bimodal particle size distribution (Figure 2), where large end tails (red arrow) indicated the presence of fine particles (<10 μm). SMPs processed at intermediate outlet temperatures (80 and 90 °C) showed smaller D_10_, D_50_, D_90_, Sauter Mean Diameter (D_[3,2]_) and Volume Mean Diameter (D_[4,3]_) than those dried at the extremes of the working temperature range (70 and 100 °C) (Table 3). These results are in agreement with the presence of shoulders (green arrow), which revealed the formation of big particles, in particular at the lower drying temperature (70 °C), where the higher moisture content may have enhanced particle aggregation. It should be highlighted that SD operating at 100 °C provided a narrower and more uniform distribution with bigger particles (Figure 2). Bista et al. [54] and Li et al. [59] obtained milk and milk protein concentrate powders, respectively, by SD (outlet temperature of 85 °C). They observed an increase in particle sizes (D_50_, D_90_ and D_[4,3]_) with processing temperature, and correlated it to heat-induced milk protein denaturation. However, this is unlikely since native whey protein structure was not dependent on outlet temperature (Figure 1).

On the other hand, PSD powders exhibited an unimodal distribution curve for all temperatures, narrower than SD, but also containing small particles (purple arrow). Wu et al. [28] dried egg white with PSD (76.6 °C) and compared it with commercial white eggs dried by SD. They observed similar differences in particle size distribution profiles, which was attributed to the absence of nozzles and rotary atomization disks of PSD. In the current study, powders obtained with PSD at 80 °C contained a higher percentage of big particles (D_50_ = 49.17 μm; D_[3,2]_ = 32.00 μm) than those processed at lower and higher temperatures. However, despite the presence of big particles at 70 and 80 °C also in PSD powders (blue arrow), an overall reduction in particle size was observed with the increment of the drying outlet temperature up to 100 °C (Table 3).

Notwithstanding, overall particle size distribution results should be handled with care, because of the intrinsic differences in dimensions and configuration between the PSD and SD pilot apparatus (a fluidized bed was available only after the PSD process).

### 3.4. Solubility, Flowability, Wettability and Compressibility

During SMP production, exposure to high temperatures may compromise the reconstitution properties of the final product (e.g., powder solubility) [45]. However, in the current study, the drying temperature did not show any significant effect on the Solubility Index (%) of SMP produced by SD and PSD. Solubility results were in agreement with Zouari et al. [60], who produced cow milk powder by SD, reporting no influence of the inlet temperature (up to 200 °C) on sample solubility (68.8–98.7%). However, when studying the influence of the drying technology, samples dried by SD were significantly more soluble (99.53 ± 0.14%) than those dried by PSD (98.43 ± 0.35%) (Table 2), but the Solubility Index was always higher than 98%, which is favorable [45].

Bulk density is a measure of the mass of powder that fills a specified volume and determines packaging, transportation and storage capacities. It is highly affected by both the drying technology used and the conditions used (e.g., solids contents, temperatures, etc.) [45,60]. Drying technology did not have a significant effect on the bulk density of SMPs dried by SD and PSD, and values ranged from 0.40 to 0.56 g/Kg (Appendix A). Nevertheless, with the increment of drying temperature, the bulk density was significantly reduced (*p* = 0.0254), achieving the minimum value at 100 °C (Table 1). In agreement with our results, low values of bulk density have previously been correlated with low moisture content [48,60]. In particular, Oliveira et al. [48] observed that bulk density increased with moisture content in goat milk powder produced by SD (up to 120 °C outlet temperature).

The flow behavior of powders can be influenced by the drying technology as well as by their composition, particle size distribution, shape or moisture content [61]. The flowability of SMP obtained by SD and PSD was expressed as the rate (g/min) at which the powder exited a rotating drum (Appendix A). The drying temperature did not have a significant effect on this parameter. However, SD samples showed lower flowability (10.68 ± 1.57 g/min) (*p* = 0.0107), indicating poorer flow properties than PSD (19.55 ± 9.82 g/min) (Table 2). Results are comparable with other studies on infant formulas [62] where higher span and increased proportion of fines resulted in lower flowability [63,64]. Therefore, Fitzpatrick et al. [64] observed that small particles in powders from different food matrices provided greater surface area with increased cohesive forces, leading to low flow properties. However, when studying compressibility (Carr Index) and Hausner ratio (H_R_) (calculated from the difference between the bulk and tapped density), neither the drying temperature nor the drying technology were significant (overall values of 33.86 ± 3.25% and 1.52 ± 0.07, respectively) and all SMPs obtained by SD and PSD were classified as distinctly cohesive (H_R_ > 1.40) [65] with very poor flowability according to Lebrun et al. [42]. High values of H_R_ and Carr Index have also been related to small particle size of powders, in agreement with flowability results [66].

The wetting time for all samples was >10 min, showing poor wettability properties, which has also been associated with the presence of small particles, as other authors reported [67,68]. Fitzpatrick et al. [67] and Wu et al. [68] observed that small particles in dried milk protein isolate difficulted water to penetrate into the powders, providing poor wettability (>60 and 36 min, respectively).

When studying the water sorption diagrams, a peak of mass increment is observed in all samples at around 50% of relative humidity (Appendix A), which likely indicates lactose crystallization [69]. However, no difference in water sorption behavior was appreciated between samples dried by PSD and SD at 70, 80, 90 and 100 °C.

Overall, all SMPs obtained showed good Solubility Indexes while PSD had somewhat better flowability. Notwithstanding, the small particle sizes of all the powders led to overall poor flow and wetting behaviors. Different strategies could be applied in order to improve these parameters, such as the addition of natural surfactants (e.g., soy lecithin) or agglomeration [45,70].

### 3.5. Scanning Electron Microscopy (SEM)

Scanning electron microscopy (SEM) was carried out to provide additional information about particle structure and surface morphology. Images obtained with SEM of SMP manufactured by SD and PSD are shown in Figure 3.

Studying the images from Figure 3, 100× magnification provided a general overview of particle distribution. In samples processed with SD, some aggregates (blue arrows in Figure 3) were present in all conditions, but to a lesser extent at 100 °C. A great number of small particles (<10 μm) were observed at low drying temperatures, which were mostly attached to the surface of larger particles. The number of fines was reduced with the temperature increment, especially at 100 °C, where particles were bigger. In PSD samples some agglomerations were present at low temperatures (70 and 80 °C). However, the sizes of the particles seemed more uniform, in agreement with the narrower particle size distribution when compared to SD samples (Figure 2). Fines were also present when drying with PSD, for all temperature conditions, that were attached as well to bigger particles’ surfaces. The presence of hollow particles was noticed in powders dried by PSD with high outlet temperatures (>90 °C) (green arrows in Figure 3), in agreement with Wu et al. [28], which may be due to a faster drying rate compared to SD [28,71].

Particles showed deep and shallow folds with wrinkles on the surface, which was similar to previous studies [60,72]. Wrinkles in SD samples were more noticeable at high temperatures (100 °C), but they were present to the same extent for all the outlet temperatures in PSD powders. Even if residence time in the chamber is short, PSD exposure to higher temperatures and fast evaporation rates described for this technology may be responsible for the formation of wrinkles even at low outlet temperatures.

## 4. Conclusions

The current work provides a first approach to understanding the impact of Pulse Spray Drying (PSD) on the physicochemical properties of skimmed milk powder (SMP). Several outlet temperatures were assessed (70, 80, 90 and 100 °C) and SMP was also obtained by traditional pilot-scale Spray Drying (SD).

The increment of outlet temperature led to reduced moisture content and bulk density, without affecting the powder solubility and native whey protein content. Although significant differences due to the drying technology were observed for yellowness and Browning Index, effects on the visual appearance of the samples were minimal. Moreover, PSD demonstrated advantages in flowability and SMP solubility was always greater than 98%. Particle size distribution and microscopy results for SD and PSD revealed the presence of fine particles influencing wetting and flow behaviors, according to the observed Hausner ratio and Carr Index values. Aggregation tendencies were observed at low drying temperatures; therefore, temperatures over 80 and 100 °C for PSD and SD, respectively, would be preferable. Even if a direct comparison between the two technologies is challenging, due to the different configurations and dimensions of the dryers, our results seem to indicate that PSD, once optimized, could offer skimmed milk powders from skim milk concentrate with an overall quality comparable to that of similar products obtained with conventional SD, but with lower energy requirements.

## Figures and Tables

**Figure 1 foods-13-00869-f001:**
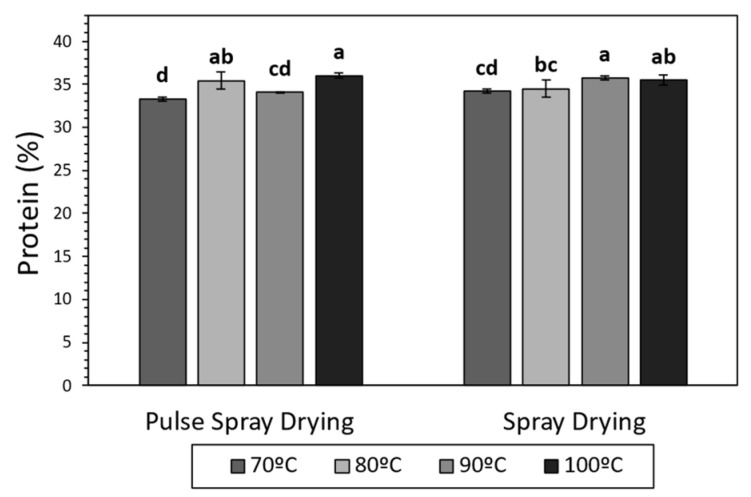
Protein content (%) of SMP obtained by Spray Drying (SD) and Pulse Spray drying (PSD) at different outlet temperatures (70, 80, 90 and 100 °C). Different letters ^(a–d)^ stand for significant differences according to Tukey’s Honest Significant Difference (HSD).

**Figure 2 foods-13-00869-f002:**
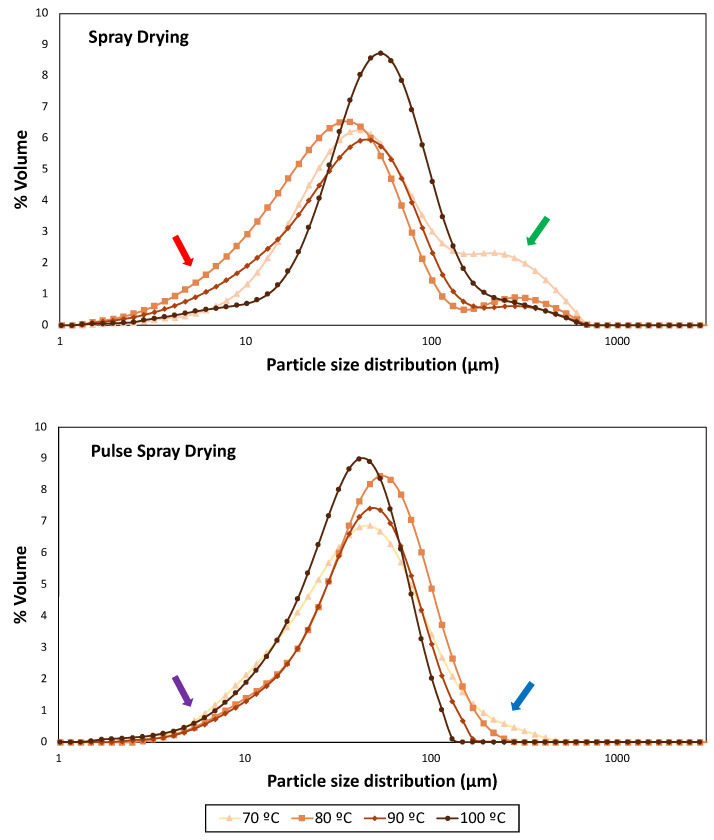
Particle size distribution profiles of samples dried at 70, 80, 90 and 100 °C with Spray Drying (SD) and Pulse Spray Drying (PSD). Red and purple arrows indicate small particles in SD and PSD samples, respectively. Green and blue arrows indicate big particles in SD and PSD samples, respectively.

**Figure 3 foods-13-00869-f003:**
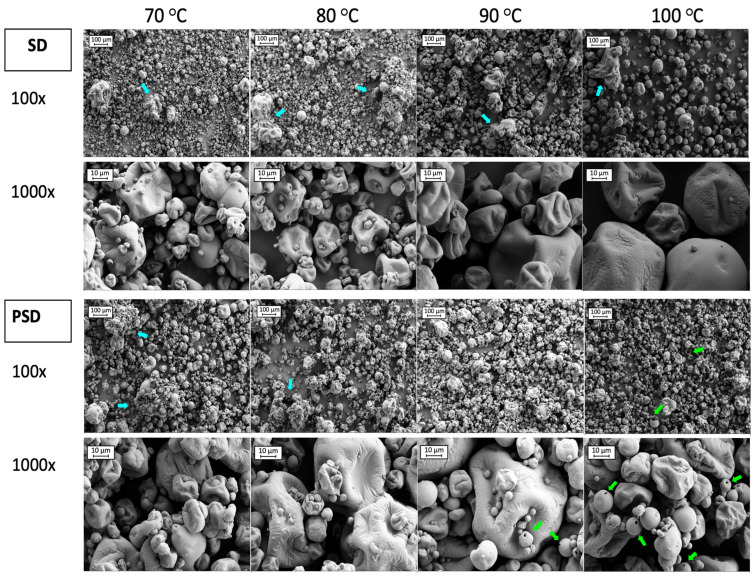
SEM images of SMPs obtained by Spray Drying (SD) and Pulse Spray Drying (PSD) at 70, 80, 90 and 100 °C. Images correspond to 100 and 1000 magnifications. Blue and green arrows indicate aggregates and hollow particles, respectively.

**Table 1 foods-13-00869-t001:** Overall values of parameters significantly affected by the drying temperature (Spray Drying: SD; Pulse Spray Drying PSD). BI: Browning Index, S_I_: Solubility Index.

	Moisture Content (%)	*b**	BI	Flowability (g/min)	S_I_ (%)
PSD	2.37 ± 1.19 ^b^	10.00 ± 0.37 ^a^	15.09 ± 0.77 ^a^	19.55 ± 9.82 ^a^	98.40 ± 0.21 ^b^
SD	2.95 ± 0.89 ^a^	9.02 ± 0.91 ^b^	12.82 ± 5.03 ^b^	10.68 ± 1.57 ^b^	99.51 ± 0.18 ^a^

Mean ± standard deviation (*n* = 16). Different letters ^(a,b)^ in the same column indicate significant differences according to Tukey’s Honest Significant Difference (HSD).

**Table 2 foods-13-00869-t002:** Effect of drying temperature on overall values of moisture content and bulk density of SMP.

Temperature (°C)	Moisture Content (%)	Bulk Density (g/mL)
70	3.21 ± 0.68 ^ab^	0.49 ± 0.04 ^a^
80	3.42 ± 1.09 ^a^	0.51 ± 0.03 ^a^
90	2.34 ± 0.58 ^bc^	0.46 ± 0.03 ^ab^
100	1.68 ± 0.94 ^c^	0.42 ± 0.04 ^b^

Mean ± standard deviation (*n* = 8). Different letters ^(a–c)^ in the same column indicate significant differences according to Tukey’s Honest Significant Difference (HSD).

**Table 3 foods-13-00869-t003:** Particle size parameters of SMPs obtained by different drying technologies (DT): Spray Drying (SD) and Pulse Spray Drying (PSD) at different drying temperatures (*n* = 4).

Temperature	DT	D_10_ (μm)	D_50_ (μm)	D_90_ (μm)	Span	D_[3,2]_ (μm)	D_[4,3]_ (μm)
70 °C	PSD	12.67 ± 1.31 ^f^	42.50 ± 5.95 ^f^	118.43 ± 30.57 ^d^	2.43 ± 0.35 ^d^	27.57 ± 3.21 ^d^	56.92 ± 12.82 ^d^
	SD	19.60 ± 0.44 ^b^	55.77 ± 0.35 ^b^	272.33 ± 0.58 ^a^	4.52 ± 0.02 ^a^	41.93 ± 0.15 ^a^	103.67 ± 0.58 ^a^
80 °C	PSD	15.47 ± 0.25 ^c^	49.17 ± 0.15 ^c^	110.33 ± 3.21 ^c^	1.93 ± 0.07 ^e^	32.00 ± 0.26 ^b^	57.63 ± 0.95 ^c^
	SD	9.02 ± 0.10 ^g^	32.63 ± 0.15 ^g^	86.40 ± 0.30 ^d^	2.37 ± 0.01 ^c^	19.47 ± 0.15 ^f^	50.87 ± 0.25 ^d^
90 °C	PSD	13.93 ± 0.25 ^d^	42.00 ± 0.15 ^d^	89.57 ± 3.21 ^d^	1.80 ± 0.07 ^ef^	29.20 ± 0.40 ^c^	49.17 ± 0.95 ^e^
	SD	9.69 ± 0.21 ^g^	37.70 ± 0.30 ^e^	105.33 ± 3.51^c^	2.54 ± 0.07 ^b^	21.20 ± 0.30 ^e^	57.00 ± 2.50 ^c^
100 °C	PSD	12.70 ± 0.10 ^e^	37.27 ± 0.06 ^ef^	75.77 ± 0.06 ^e^	1.69 ± 0.01 ^f^	24.47 ± 0.25 ^d^	41.33 ± 0.06 ^e^
	SD	25.97 ± 0.15 ^a^	57.90 ± 0.10 ^a^	145.00 ± 1.00 ^b^	2.06 ± 0.01 ^d^	43.53 ± 0.15 ^a^	79.03 ± 0.25 ^b^

Mean ± standard deviation. Different letters ^(a–g)^ in the same column indicate significant differences according to Tukey’s Honest Significant Difference (HSD).

## Data Availability

The original contributions presented in the study are included in the article/Appendix A, further inquiries can be directed to the corresponding author.

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
