# Peer review of "Pulse Spray Drying for Bovine Skimmed Milk Powder Production"

_foods, 2024, doi:10.3390/foods13060869_

Round 1
Reviewer 1 Report
Comments and Suggestions for Authors
The manuscript entitled "Pulse Spray Drying for Bovine Skimmed Milk Powder production" shows interesting results, worth of publication. However, due to many shortcomings this manuscript requires major revision.
The paper is written quite well, the methods used are sound and but the data presented does not support the discussion made. The main objective is not answered. By the way, if a major rewrite of this manuscript were performed, it should be acceptable.
Comments on the Quality of English Language
The manuscript entitled "Pulse Spray Drying for Bovine Skimmed Milk Powder production" shows interesting results, worth of publication. However, due to many shortcomings this manuscript requires major revision.
The paper is written quite well, the methods used are sound and but the data presented does not support the discussion made. The main objective is not answered. By the way, if a major rewrite of this manuscript were performed, it should be acceptable.
Author Response
The manuscript entitled "Pulse Spray Drying for Bovine Skimmed Milk Powder production" shows interesting results, worth of publication. However, due to many shortcomings this manuscript requires major revision.
The paper is written quite well, the methods used are sound and but the data presented does not support the discussion made. The main objective is not answered. By the way, if a major rewrite of this manuscript were performed, it should be acceptable.
Thank you for your comments and suggestions. We improved the discussion and tried to delete the shortcomings.
Reviewer 2 Report
Comments and Suggestions for Authors
This research paper has a reasonable structure and a smooth writing. Rich results are provided in the paper and can be read directly from various tables and charts. However, the authors gave less space in the discussion section. A large amount of practice is acceptable, and in fact, it is important to discuss the changes caused by practice. The authors need to deepen the discussion of the topic in the paper. In addition, there are the following recommendations to consider.
Point 1: The annotation of the reference [12] is not represented in the article.
Point 2: Line 106, What is the reason why the inlet temperature is set to "160,180,190 and 210℃", and will it have an impact on the experimental results?
Point 3: Line 135, with a space in the middle of "10-12", check the entire text to ensure that the format is consistent.
Point 4: Line 171, There are two ")" , one needs to be deleted, Please check the entire text to ensure consistent formatting.
Point 5: Line 205, The explanation of several parameters has been mentioned earlier, and there is no need to reinterpret the parameter definition here, so it can be deleted.Please check the entire text to ensure consistent formatting.
Point 6: In Table 3, the unit of temperature does not correspond with the unit position of water content and volume density.
Point 7: Line 205, the solubility index situation does not coincide with the solubility index situation in line 143. Please check the full text and make corrections to ensure consistency in the format.
Point 8: Line 249-269, In the process of temperature increase to 100℃, the particle size change law of SD and PSD is different, what causes it.
Author Response
This research paper has a reasonable structure and a smooth writing. Rich results are provided in the paper and can be read directly from various tables and charts. However, the authors gave less space in the discussion section. A large amount of practice is acceptable, and in fact, it is important to discuss the changes caused by practice. The authors need to deepen the discussion of the topic in the paper. In addition, there are the following recommendations to consider.
Thanks a lot for your comments and suggestions. We have improved the discussion and replied to the several points (see below).
Point 1: The annotation of the reference [12] is not represented in the article.
Corrected.
Point 2: Line 106, What is the reason why the inlet temperature is set to "160,180,190 and 210℃", and will it have an impact on the experimental results?
PSD is a drying emerging technology that has not been optimized in skim milk. Thus, in order to study its impact on the product quality, a wide range of outlet temperatures was selected (from 70 ºC to 100 ºC), including temperatures below and over than those normally applied in dairy industry. The same outlet temperatures were selected for SD drying, that were adjusted by applying different inlet temperatures inside the drying chamber (from 160 ºC to 210 ºC). The exposure of higher temperatures (e.g., 100 ºC) may lead to undesirable changes in the products. However, in the current study it was demonstrated that parameters (e.g., color, whey proteins, solubility…) were not impacted by temperature, probably due to the short residence time of the milk inside the drying chamber.
Point 3: Line 135, with a space in the middle of "10-12", check the entire text to ensure that the format is consistent.
Corrected.
Point 4: Line 171, There are two ")" , one needs to be deleted, Please check the entire text to ensure consistent formatting.
Corrected.
Point 5: Line 205, The explanation of several parameters has been mentioned earlier, and there is no need to reinterpret the parameter definition here, so it can be deleted. Please check the entire text to ensure consistent formatting.
Entire text checked.
Point 6: In Table 3, the unit of temperature does not correspond with the unit position of water content and volume density.
Corrected.
Point 7: Line 205, the solubility index situation does not coincide with the solubility index situation in line 143. Please check the full text and make corrections to ensure consistency in the format.
Corrected.
Point 8: Line 249-269, In the process of temperature increase to 100℃, the particle size change law of SD and PSD is different, what causes it.
The effect of drying temperature on particle size distribution varied depending on the drying technology. Further studies should be conducted to study better the exact reason of these disparities, since very little literature is available about PSD applied on food matrices. However, it should be noted that, in the current study, this could be explained by differences between PSD and SD in the drying mechanism, as well as in the equipment dimensions and configurations (e.g., fluidized bed only present in PSD process, higher inlet temperatures in PSD…), that strongly influenced particle size distributions. A sentence mentioning this has been added in the manuscript.
Reviewer 3 Report
Comments and Suggestions for Authors
María Romo , Doll Chutani , Dinar Fartdinov , Ram Raj Panthi , Nooshin Vahedikia , Massimo Castellari , Xa-4 vier Felipe , and Eoin G. Murphy
Pulse Spray Drying for Bovine Skimmed Milk Powder production
Rewier comments: major revisions
The paper reports a study on Pulse Spray Drying (PSD) as a sustainable means of skimmed milk powder (SMP) production.
The manuscript can be a good contribution. However, several parts have to be improved especially in Results and discussion and Materials and methods paragraphs.
The authors created a big list of references but in the text they did not compare efficiently their study with them.
In these reasons the manuscript need the following modifications:
Introduction
In the introduction the author could explain deeply the application of PSD to food matrices before the application to skimmed milk powder.
Line 78: Please specify in details the drying conditions
Line 81: Please specify what is SMP
Materials and methods
2.2. Milk sample preparation
Please, add information on why this specific experiment was selected or add a reference
2.6. Powder physical properties
Lines 141-144: Please, rewrite the Insolubility Index and Solubility Index definition. It is not clear.
Results and discussion
The authors have to improve al lot several part of discussion. They have to discuss deeply their results comparing them with those obtained in other studies. It is not only sufficient to cite them.
3.1. Protein and moisture content
Lines 216-219: the authors said that they have quantified the alfa and beta lactalbumin but they did not explain in the text any result. Please improve this part in the manuscript.
3.3. Particle size and distribution
Lines 257-260: Please, the author should do a comparison with those found in other studies.
3.4. Solubility, flowability, wettability and compressibility 273
Line 277: Please, the author should do a comparison with those found in the study of Zouari et al.
Line 305-307: also here, Please, the author should do a comparison with those found in other studies.
References
Please, Revise the references (number DOI, authors, number pages or issues, etc.)
Author Response
The paper reports a study on Pulse Spray Drying (PSD) as a sustainable means of skimmed milk powder (SMP) production. The manuscript can be a good contribution. However, several parts have to be improved especially in Results and discussion and Materials and methods paragraphs.The authors created a big list of references but in the text they did not compare efficiently their study with them. In these reasons the manuscript need the following modifications:
Thanks a lot for your comments and suggestions. We have improved the discussion and replied to the several points (see below).
Introduction
In the introduction the author could explain deeply the application of PSD to food matrices before the application to skimmed milk powder.
PSD has been mostly applied to inorganic matrices (e.g., zinc oxide, wood, paper…). To the best of our knowledge, there is a lack of studies of PSD applied on food matrices, only the one on white egg by Wu et al. (2014) (lines 77-80).
Line 78: Please specify in details the drying conditions
Added.
Line 81: Please specify what is SMP
Added.
Materials and methods
2.2. Milk sample preparation
Please, add information on why this specific experiment was selected or add a reference
Reference added.
2.6. Powder physical properties
Lines 141-144: Please, rewrite the Insolubility Index and Solubility Index definition. It is not clear.
Corrected.
Results and discussion
The authors have to improve al lot several part of discussion. They have to discuss deeply their results comparing them with those obtained in other studies. It is not only sufficient to cite them.
Discussion improved.
3.1. Protein and moisture content
Lines 216-219: the authors said that they have quantified the alfa and beta lactalbumin but they did not explain in the text any result. Please improve this part in the manuscript.
Part improved; values added.
3.3. Particle size and distribution
Lines 257-260: Please, the author should do a comparison with those found in other studies.
Part improved; comparison done.
3.4. Solubility, flowability, wettability and compressibility 273
Line 277: Please, the author should do a comparison with those found in the study of Zouari et al.
Part improved; comparison done.
Line 305-307: also here, Please, the author should do a comparison with those found in other studies.
Part improved; comparison done.
References
Please, Revise the references (number DOI, authors, number pages or issues, etc.)
References reviewed.